# Mutual induced-fit mechanism drives binding between intrinsically disordered Bim and cryptic binding site of Bcl-xL

Gert-Jan Bekker [1✉], Mitsugu Araki [2], Kanji Oshima[3], Yasushi Okuno [2] & Narutoshi Kamiya [4✉]

The intrinsically disordered region (IDR) of Bim binds to the flexible cryptic site of Bcl-xL, a pro-survival protein involved in cancer progression that plays an important role in initiating apoptosis. However, their binding mechanism has not yet been elucidated. We have applied our dynamic docking protocol, which correctly reproduced both the IDR properties of Bim and the native bound configuration, as well as suggesting other stable/meta-stable binding configurations and revealed the binding pathway. Although the cryptic site of Bcl-xL is predominantly in a closed conformation, initial binding of Bim in an encounter configuration leads to mutual induced-fit binding, where both molecules adapt to each other; Bcl-xL transitions to an open state as Bim folds from a disordered to an α-helical conformation while the two molecules bind each other. Finally, our data provides new avenues to develop novel drugs by targeting newly discovered stable conformations of Bcl-xL.

[1] Institute for Protein Research, Osaka University, 3-2 Yamadaoka, Suita, Osaka 565-0871, Japan. [2] Graduate School of Medicine, Kyoto University, 53 Shogoin-Kawaharacho, Sakyo-ku, Kyoto 606-8507, Japan. [3] Bio-Pharma Research Laboratories, KANEKA CORPORATION, 1-8 Miyamae-cho, Takasago-cho, Takasago, Hyogo 676-8688, Japan. [4] Graduate School of Information Science, University of Hyogo, 7-1-28 Minatojima Minami-machi, Chuo-ku, Kobe, Hyogo 650-0047, Japan. ✉email: gertjan.bekker@protein.osaka-u.ac.jp; n.kamiya@sim.u-hyogo.ac.jp

B cell lymphoma 2 (Bcl-2) family proteins are important regulators in the apoptotic pathway inside cells (Fig. S1)[1,2]. Bcl-xL, along with Bcl-2, Bcl-w, Bfl-1, and Mcl-1 are pro-survival proteins part of the Bcl-2 family[3] that interact with pro-apoptotic proteins such as effector proteins Bax and Bak[4], upstream initiator proteins within the pathway, or Bcl-2 homology 3 only (BH3-only) proteins such as Bcl-2 interacting mediator (Bim), Puma, Bad and Bmf[5,6]. Bcl-xL suppresses these pro-apoptotic proteins by binding to their BH3-motif, inhibiting their apoptotic signal[2]. These BH3-only proteins are intrinsically disordered proteins (IDPs), and the intrinsically disordered region (IDR) of the BH3-motif (L-XXX-[GA]-D) undergoes a conformational change upon binding to their pro-survival partner molecule[7]. In complex with their partner molecules such as Bim, the structure of the BH3-motif forms an α-helical conformation, with their hydrophobic residues interacting inside hydrophobic grooves on the surface of their partners from the Bcl-2 family, forming a stable protein-protein complex. Four of these sub-pockets of Bcl-xL that bind hydrophobic residues from the BH3-motif have been labeled as P1-P4 (Fig. 1a)[4]. Because over-expression of Bcl-xL is one of the hallmarks of cancer[8], Bcl-xL is considered to be an important drug target, whereby an BH3-mimic inhibitor that binds to the BH3 binding site would be able to prevent Bcl-xL from inhibiting apoptotic signals, halting the growth of cancer cells.

Structurally, Bcl-xL in complex with natural/unnatural proteins/peptides, chemical compounds, as well as its apo-form, has been well-studied by X-ray crystallography and nuclear magnetic resonance (NMR) spectroscopy[2,9]. Bcl-xL is an all α-helical protein consisting of eight α-helices (α1-α8 in Fig. 1a), and this fold is conserved among Bcl-2 family proteins. The interface of Bcl-xL, which can bind to the BH3 motif of pro-apoptotic proteins, contains a large hydrophobic groove formed by α2-α4 with the core helix α5[10]. The binding site on Bcl-xL is a so-called cryptic binding site. A cryptic binding site only exposes their binding site upon binding of a ligand, or only transiently in the unbound state[11,12]. This also means that cryptic binding sites can be highly flexible[13,14], some to such a degree that they could be considered to be weak IDRs. A high resolution X-ray structure of Bcl-xL in complex with a peptide fragment of Bim inside the cryptic binding site that adopts an α-helical conformation has been reported (Fig. 1b)[15]. Several drugs that bind to the same binding site as Bim have been rationally designed based on known crystal structures, which has resulted in the discovery of several medium-sized inhibitor molecules such as ABT-737[16] and WEHI-539[17]. Although both peptide-bound and chemical compound-bound Bcl-xL structures have been solved, the binding mechanism between Bcl-xL and Bim remains unclear. In particular, understanding the molecular recognition mechanism between Bcl-xL and peptide fragments such as Bim is challenging, because the region of Bim that binds Bcl-xL is an IDR, while the cryptic binding site of Bcl-xL is also highly flexible. To study IDRs, advanced computational and theoretical approaches are required to study the structure and dynamics of monomeric and higher order assemblies[18]. In addition, in structures solved by AlphaFold[19], IDRs tend to correspond with regions of very low confidence[20], meaning that advanced computational techniques are required to accurately model IDRs.

There are two major conformational change mechanisms in a receptor protein that describe the binding to a ligand[21]. The first is the conformational selection model, where the conformational ensemble of the receptor consists of a large number of conformations, but binding is only attempted if the receptor is in a bound-like conformation. The second one is the induced-fit mechanism, where the ligand binds the receptor in an arbitrary conformation, forming an encounter complex that causes the

receptor to change its conformation to the bound-like conformation. Although molecular recognition between chemical compounds and globular proteins tends to follow the conformational selection model, IDPs and IDRs tend to follow the induced-fit model, specifically coupled folding and binding[22]. As for recognition of larger systems such as protein-protein or protein-peptide, we previously performed dynamic docking simulations between an antibody and an antigen, where we observed a conformational selection mechanism in both molecules, which lead to a mutual population-shift in the conformational ensembles of both molecules[23]. However, due to the computational requirements of docking between molecules that contain IDRs, computational studies of the binding mechanism of IDRs and IDPs, especially to highly flexible pockets such as cryptic binding sites, are still quite rare.

Docking using molecular dynamics (MD) simulations, called dynamic docking, can be used to explore binding configurations between receptor proteins and their ligands[24]. We have developed a dynamic docking implementation based on multicanonical molecular dynamics (McMD, see Note S1 for an explanation of the McMD theory)[25], which we have applied to a number of cases from small-molecule ligands to medium-sized ligands including peptides around 10 residues[23,26–32]. We have applied McMD[33,34] simulations to the conformational sampling of proteins and peptides[35,36] and the loop structure prediction of an antibody[37]. With McMD, the bias is correlated with the temperature, enabling McMD simulations to adaptively modulate the bias given the density of states. Thus, the potential energy surface functions as a reaction coordinate, which does not depend on any prior knowledge (e.g., native receptor-ligand complex). The

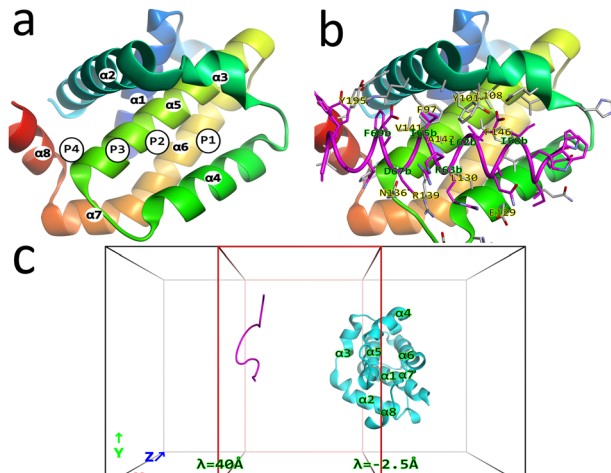

**Fig. 1 Structural overview of Bcl-xL and Bim with the computational system. a** Structure of Bcl-xL and location of the binding sites P1-P4, with the secondary structure elements α1-α8 labeled. **b** Complex structure of Bcl-xL (colored with a blue-red gradient from N- to C-terminal) and Bim (magenta), with contacting residues shown and labeled, with residues of Bim suffixed with a "**b**". **c** One of the initial structures ($N = 30$) of the computational system at the start of the simulations after the initial high-temperature simulations, with Bcl-xL in cyan and Bim with sequence CH$_3$CO-IWIAQE**LRRIGD**EFNAYY-NH$_2$ in magenta (BH3-motif in bold). The center-of-mass (COM) of Bim is restrained to stay inside the red box along the x-axis. This is similar to the cylinder used in previous works with the axis $\lambda$ corresponding to the x-axis, and $\lambda$ ranging from $-2.5$ Å to 40.0 Å, where a force constant of 10 kcal/mol was used when the COM of Bim leaves this range along the axis, while no force was applied when the COM leaves the box perpendicular to the axis, thus periodic boundary crossings along the y- and z-axis were allowed. The images were produced using Molmil[64], a WebGL based molecular viewer developed by PDBj[48–50].

canonical ensemble at any given temperature, which is one of the physico-chemically acceptable ensembles, can be generated from the multicanonical ensemble by using a reweighting procedure. The free energy landscape (FEL), which governs the thermo-dynamic properties of a system, can then be obtained by mapping the reweighted structural ensemble onto a reaction coordinate such as a binding path or onto one or more principal components obtained by Principal Component Analysis (PCA)[35,38]. Analysis of the FEL then uncovers the stable bound complexes as sampled by the McMD simulation. From the stable structures, a ligand binding path can be produced by tracing nearest neighbor structures from the multicanonical ensemble[25,28,39].

We previously used McMD-based dynamic docking to analyze the binding mechanism between Bcl-xL and two medium-sized compounds, WEHI-539 and ABT-737[29]. In this work, we are taking a closer look at the binding between Bcl-xL and its natural ligand, Bim, which forms an α-helical structure upon binding to Bcl-xL (Fig. 1). The McMD-based dynamic docking algorithm was able to successfully predict the native bound structure and reproduce both the IDR properties of Bim in isolation, as well as a strong helical conformation in the presence of Bcl-xL. To elucidate the binding mechanism, analysis of the binding pathway revealed that binding appears to be initiated via the C-terminal region of Bim, as the corresponding binding site on Bcl-xL (P4) remains accessible, while the binding site for the N-terminal region (P1 and P2) is very flexible and is often in a closed conformation that cannot bind fully to Bim. As Bcl-xL attains an open conformation, more of Bim binds to Bcl-xL as it folds into a helical conformation. With our previous simulations we demonstrated how medium-sized inhibitors bind to the cryptic binding site of Bcl-xL, and here we show how its natural ligand Bim binds to the same highly flexible binding site and further-more demonstrate how each ligand influences the conformational ensemble of Bcl-xL as it binds.

## Results

Since the BH3 domain of Bim is an IDR, we first decided to execute McMD simulations on the peptide (18-mer, see Methods section or Fig. 1 for the sequence) in isolation, to verify that the force field was appropriate for IDPs/IDRs. As we recently also performed simulations of a membrane-embedded GPCR mole-cule using the AMBER ff14 force field[40], we wanted to compare this force field against the AMBER ff99SB-ILDN force field that we have used in the past[23,30]. For each force field, we performed 150 ns of pre-run, followed by 250 ns of production-run sampling for each parallel McMD trajectory ($N = 30$) and this for each force field. The obtained multicanonical distributions are shown in Fig. S2, with the obtained FELs in Fig. S3. The statistics regarding picked representative structures from the FELs are shown in Table S1 and S2 for the ff14 and ff99SB-ILDN force field, respectively. Similar to our picking method for binding configurations[28,39,41], the picking method for single-chain pro-tein/peptide conformations also uses K-means clustering on the PC coordinates (Table S1). The top-ranking structure, i.e., the energetically most stable one, using the ff14 force field was very similar to the experimental structure in complex with Bcl-xL (Table S1), while the similarity of the top-ranking structure using the ff99SB-ILDN force field was instead very low (Table S2). Furthermore, the number of structures within the 2.5 kcal/mol cutoff was considerably larger for the ff99SB-ILDN force field than the ff14 force field. Taken together, the ff99SB-ILDN force field produced a much more diverse conformational ensemble of Bim at 300 K without preference for a specific structure, speci-fically, no preference for the conformation observed in the complex structure (PDB ID 4QVF)[15]. Thus, we chose to perform the

docking simulations with ff99SB-ILDN, as this force field pro-duced a wide structural ensemble that is expected of IDPs/IDRs.

Settling on the ff99SB-ILDN force field, we subsequently pre-pared the docking system, using the apo Bcl-xL receptor structure that we used in our previous work[29] and the ligand Bim peptide, as shown in Fig. 1b, and subsequently executed the dynamic docking simulations. During the first iteration of the McMD simulations, the simulation temperature was set to $T_{high} = 700$ K, resulting in unfolding and randomization of the Bim structure (Fig. 1c). After an 800 ns pre-run for each parallel McMD trajec-tory ($N = 30$, Table S3), the production run was executed, with 1 μs per trajectory, producing a multicanonical ensemble con-sisting of $6.0 \times 10^6$ structures. The flat potential energy distribu-tion obtained from the production run is shown in Fig. S4, with the reweighting distributions (Eq. S5) for $T$ at 300 K, 500 K and 700 K. Projecting the reweighted ensembles (300 K) onto the first two principal axes obtained via PCA, we obtain a FEL (see Methods) as shown in Fig. 2a, where the experimental structure is located close to the global minimum. To study the stable binding configurations in greater detail, we picked representative struc-tures from the multicanonical ensemble, where in Fig. 2b the representative structures $\mathbf{r}$ of the clusters $k$ that have a CFE of less than the 2.5 kcal/mol have been mapped onto the FEL. Fig. S5 shows the structures for these representative configurations, where Table 1 lists several statistics of them, where these struc-tures are ranked by the free energy contribution of the corre-sponding cluster. In addition, analysis results using 25%, 50% and 75% of the simulations from the production run are shown in Table S4. Here, the top-ranking structure in each subsection is consistently the native configuration, indicating that the docking simulations have quickly converged.

The top-ranking structure $\mathbf{r}_1$ has roughly the same binding configuration as the experimental structure, but has a relatively low R(native)-value[27,42] of 0.66 and a root-mean-square deviation (RMSD) of 2.79 Å, suggesting that the contacts differ somewhat even though the location on the FEL is nearby the experimental structure. Comparing the experimental structure (Fig. 2c) with $\mathbf{r}_1$ (Fig. 2d), shows that α3 of Bcl-xL is in a much more elongated, coiled-coil conformation in the experimental structure, while it mostly is in an α-helical conformation in $\mathbf{r}_1$. Furthermore, Phe105 is buried inside Bcl-xL, packing against α5 in $\mathbf{r}_1$, while in the experimental structure it is facing outwards in a conformation that would be unstable in explicit water. In the experimental structure, Leu108 is positioned where Phe105 is in $\mathbf{r}_1$, while Leu108 in $\mathbf{r}_1$ interacts with Val126. Due to crystal contacts made under experimental conditions (Fig. S6), Phe105 was exposed to the surface, but in solution, such a conformation would not be stable, thus we are observing Phe105 being buried in $\mathbf{r}_1$, and the conformation of α3 changed to adapt to this energetic require-ment. The result is that in $\mathbf{r}_1$, not only Phe105 becomes buried, but that the entire conformation of α3 is changed, including movement of sidechain atoms, reducing the R(native)-score, while Bim is still roughly positioned similarly to the experimental structure. Thus, the presence of crystal packing gave rise to an unrealistic Phe105 conformation that would not have been stable without crystal packing, which in turn lead to large conforma-tional changes of α3. Since the structures otherwise largely match, it is safe to assume that $\mathbf{r}_1$ corresponds to the native structure in a solvated environment. On the other hand, the other meta-stable configurations consist of a mixture of partially folded Bim con-formations, as well as non-native configurations (Fig. S5 and Table 1). Here, $\mathbf{r}_3$, $\mathbf{r}_5$, $\mathbf{r}_7$ and $\mathbf{r}_8$ are in a bound configuration similar to the native configuration, but have either or both of the termini of Bim unfolded to varying degrees. For $\mathbf{r}_2$, Leu62b is bound nearby site P2, with Tyr73b located close to P4, while the peptide is otherwise in an unfolded non-native state and Bcl-xL in

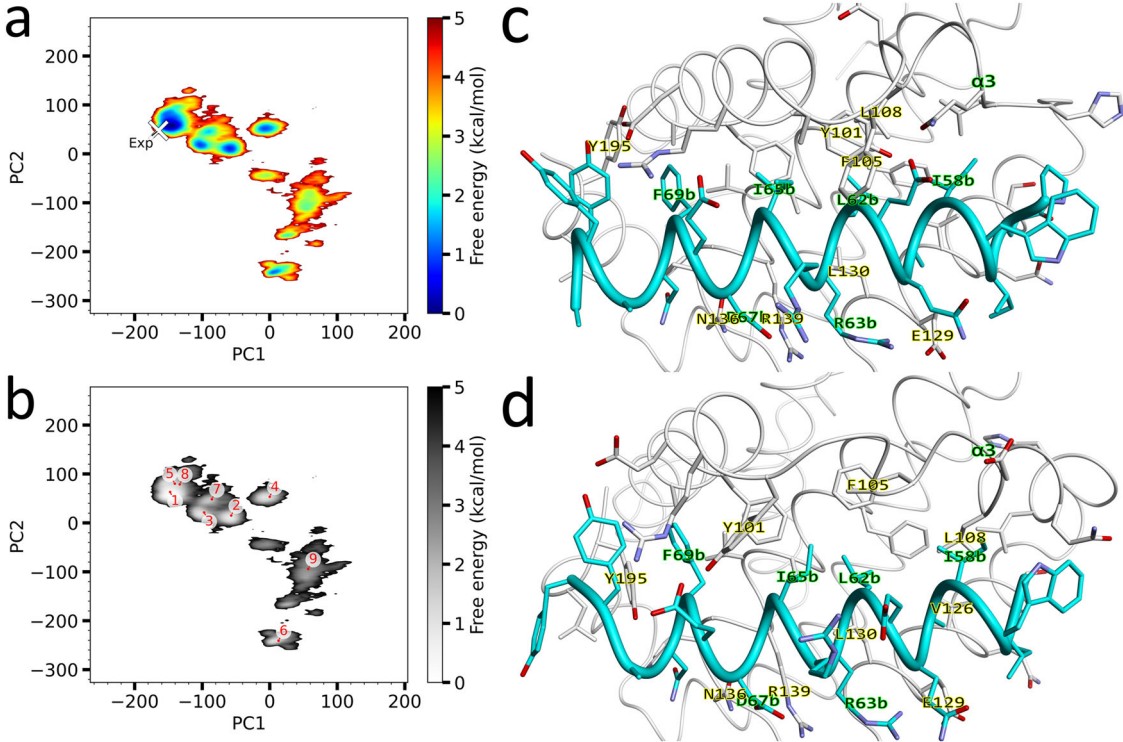

**Fig. 2 Dynamic docking results between Bcl-xL and Bim. a** Free energy landscape (FEL) of binding configurations along the first and the second principal components (PC1 and PC2, respectively), obtained from the reweighted (300 K) multicanonical ensemble, with the contour value in kcal/mol. The location of the experimental structure (PDB ID 4QVF) is indicated by a white-colored X with the text "Exp". **b** FEL with the locations of the representative structures $r_k$ in red, with the characteristics of these structures listed in Table 1 and the structures themselves shown in Fig. S5. **c** Experimental structure with PDB ID 4QVF, and **d** top-ranking structure $r_1$ (energetically most stable structure). Shown are Bcl-xL in white and Bim in cyan, along with the sidechain of interacting residues and their labels.

**Table 1 Stable binding configurations obtained from the reweighted (300 K) multicanonical ensemble (up to 2.5 kcal/mol)[a].**

|        | CFE (kcal/mol) | PC1     | PC2     | PCA FE (kcal/mol) | RASA | $R$ (native)-value | RMSD (Å) | λ (Å) |
|--------|----------------|---------|---------|-------------------|------|--------------------|----------|-------|
| $r_1$  | 0.00           | −147.55 | 63.32   | 0.18              | 0.53 | 0.660              | 2.79     | 5.52  |
| $r_2$  | 0.98           | −56.98  | 15.13   | 0.89              | 0.63 | 0.294              | 6.80     | 7.46  |
| $r_3$  | 1.15           | −97.1   | 21.39   | 1.19              | 0.55 | 0.558              | 8.23     | 7.58  |
| $r_4$  | 1.18           | 0.00    | 53.51   | 1.15              | 0.65 | 0.127              | 12.69    | 10.80 |
| $r_5$  | 1.30           | −140.63 | 80.04   | 1.21              | 0.59 | 0.565              | 5.58     | 5.06  |
| $r_6$  | 1.50           | 13.11   | −240.94 | 1.32              | 0.54 | 0.002              | 16.00    | 7.56  |
| $r_7$  | 1.99           | −85.77  | 48.27   | 1.76              | 0.57 | 0.425              | 6.24     | 7.78  |
| $r_8$  | 2.28           | −133.19 | 78.80   | 1.54              | 0.59 | 0.609              | 6.09     | 4.19  |
| $r_9$  | 2.34           | 56.93   | −94.28  | 2.57              | 0.68 | 0.109              | 17.24    | 9.97  |
| Exp    | -              | −164.72 | 47.80   | 3.94              | 0.53 | 1.000              | 0.00     | 5.94  |

[a]Characteristics for representative structures $r_k$ (from McMD). Shown are, the cluster free energy (CFE) value in kcal/mol of the corresponding cluster $k$, the first two principal components (PC1, PC2), the free energy in kcal/mol of the point (PC1, PC2) on the FEL (PCA FE) in Fig. 2a, the fraction of the relative accessible surface area (RASA) of the Bim peptide, the R(native)-value with respect to the experimental structure, the RMSD in Å of the heavy peptide atoms with respect to the experimental structure, and the λ-value (Fig. 1). Analysis results using 25%, 50% and 75% of the simulations from the production run are shown in Table S4.

a somewhat closed state, similar to the apo conformation. Finally, $r_4$, $r_6$ and $r_9$ are configurations that have Bim bound in a non-native configuration to Bcl-xL, with Bim in a non-helical conformation.

We have also performed canonical MD simulations at 300 K using the structures $r_k$ to refine them (to produce the equilibrated structures $q_k$) and at 400 K (to analyze their relative stability) for the structures with a CFE of <2.5 kcal/mol. The statistics for these structures are listed in Table 2, with the equilibrated binding configurations shown in Fig. S7. The top-ranking configuration $r_1$ has the highest stability, however some of binding configurations within 2.5 kcal/mol exhibit similar stabilities. The measured

stability for the top-ranking configuration is however lower compared to that of the small compounds we previously analyzed[26–29,31]. Analyzing the per-residue $R$-value of Bim in each binding configuration (Table S5), we can see that the terminal residues are particularly weak, while the core residues maintain their contacts much better for $r_1$. Given that the configurations $r_3$, $r_5$, $r_7$ and $r_8$ are bound in a similar manner as $r_1$, except for changes in the termini, one could consider all these separate configurations as a single one, with the termini in a semi-disordered state. The configuration $r_2$ however has a completely different per-residue stability profile, with the N-terminal residues binding strongly, weakening towards the core, while the

**Table 2 Refined binding configurations produced by canonical MD simulations and stability estimation of $r_k$ using R-value analysis[a].**

|  | PC1 | PC2 | PCA FE (kcal/mol) | RASA | R (native)-value | RMSD (Å) | λ (Å) | *R*-value 300 K | *R*-value 400 K |
|---|---|---|---|---|---|---|---|---|---|
| $q_1$ | −153.06 | 61.52 | 0.40 | 0.56 | 0.686 | 2.61 | 5.43 | 0.937 (0.046) | 0.848 (0.059) |
| $q_2$ | −53.75 | 5.89 | 1.22 | 0.62 | 0.290 | 6.97 | 7.68 | 0.952 (0.047) | 0.641 (0.179) |
| $q_3$ | −95.71 | 16.53 | 1.26 | 0.53 | 0.580 | 8.22 | 7.78 | 0.924 (0.057) | 0.832 (0.072) |
| $q_4$ | −6.31 | 47.34 | 1.45 | 0.63 | 0.146 | 12.44 | 10.37 | 0.981 (0.013) | 0.760 (0.232) |
| $q_5$ | −134.58 | 89.67 | 2.21 | 0.61 | 0.570 | 6.00 | 5.08 | 0.909 (0.043) | 0.769 (0.067) |
| $q_6$ | 12.11 | −242.38 | 1.38 | 0.53 | 0.030 | 16.20 | 7.32 | 0.967 (0.032) | 0.894 (0.075) |
| $q_7$ | −85.72 | 43.62 | 1.95 | 0.57 | 0.424 | 6.38 | 7.57 | 0.968 (0.021) | 0.580 (0.097) |
| $q_8$ | −133.94 | 78.67 | 1.49 | 0.55 | 0.594 | 5.85 | 3.84 | 0.915 (0.033) | 0.750 (0.131) |
| $q_9$ | 55.94 | −98.18 | 2.54 | 0.69 | 0.112 | 17.41 | 10.11 | 0.864 (0.076) | 0.629 (0.178) |

[a]Characteristics for representative structures $q_k$. Shown are, the first two principal components (PC1, PC2), the free energy in kcal/mol of the point (PC1, PC2) on the FEL (PCA FE) in Fig. 2a, the fraction of the relative accessible surface area (RASA) of the Bim peptide, the R(native)-value with respect to the experimental structure, the RMSD in Å of the heavy peptide atoms with respect to the experimental structure, the λ-value (Fig. 1), the average *R*-value (with respect to the initial structure $r_k$) of the final 40 ns over 10 parallel trajectories at 300 K (with standard deviation), and the average *R*-value of the final 40 ns over 10 parallel trajectories at 400 K (with standard deviation).

C-terminal residues show a similar stability as the native configurations. Finally, since the other configurations bind in different manners, this leads to different per-residue *R*-value profiles. Notably, however, is $r_6$, which displays a high stability similar to the native bound configuration $r_1$, even though it binds in a non-native, closed state and thus might be an interesting configuration to mimic to develop novel inhibitor compounds.

To analyze the binding pathway, we used our pathing algorithm to generate a set of structures along the *x*-axis (or λ, Fig. 1c) starting from $q_1$. Table S6 lists the structural similarities of the picked structures between neighboring windows. As the peptide dissociates, the picked representative for each window has both fewer representative structures, and in some cases a somewhat low similarity score with the previous window. Here, the similarity is measured by intermolecular contact similarity (*R*-value) and molecular displacement (RMSD). In particular, window 3 has a relatively low similarity score (R = 0.750, RMSD = 7.23 Å), while the representative structure picked in window 6 also has quite a low similarity score (R = 0.544, RMSD = 5.56 Å), suggesting that there are high energy barriers in these regions. Inspecting the structures, the binding pathway produced by our algorithm corresponds to one where binding appears to be initiated at the C-terminus of Bim to P4, where as more of Bim binds, the peptide slowly folds into an α-helical conformation until its N-terminal binds (Fig. S8, see the Discussion section for a detailed description of the binding mechanism).

In order to analyze the pocket size of Bcl-xL along the cryptic pocket using the multicanonical ensemble to understand why binding is initiated at P4, we calculated the distribution of distances between a the Cα atoms of a set of residues (Leu112-Ser122, Leu108-Val126, Phe105-Leu130, Tyr101-Arg139 and Glu96-Asn136, Fig. S9) along the pocket, defined as d0-d4, of the multicanonical ensemble and reweighted those distributions to 300 K, 500 K and 700 K, to explore how the pocket's shape and dynamics influence the binding of Bim (Fig. S10). Notably, at room temperature for $d1$ (near P1) and $d2$ (near P2), a multi-peak distribution can be observed, while the other locations only show a single peaked distribution. Inspecting the structures of $r_1$, $r_6$ and $r_8$, which have distances at $d1$ of 9.22, 7.25 and 11.19 Å, respectively, and distances at $d2$ of 12.32, 10.66 and 13.57 Å respectively (Table S7), shows that the conformation of α3 and α4 differs between them, with $r_6$ clearly in a closed state, with $r_1$ and $r_8$ in a more open state, based on their backbone structures (Fig. S11). Although the pocket size of Bcl-xL in both $r_1$ and $r_8$ appears to be quite similar given their backbone structure, the Phe105 sidechain flip in $r_8$ causes a partial block of the pocket,

preventing the N-terminal of Bim from binding. However, it should be noted though that the values of both $d1$ and $d2$ for $r_1$ are located in minor peaks, suggesting that the conformation of Bcl-xL in $r_1$ is only stable with Bim bound in its native conformation. When looking at the 500 K distributions, we also observe that for $d1$ the majority of the structures now have a shorter distance, meaning structures that are predominantly in a closed conformation. On the other hand, the pocket size on the C-terminal side of Bim (i.e., $d3$ and $d4$) is much more consistent, with only a single peak observed, which are maintained, although being weaker, at higher temperatures. Finally, at 700 K, the distributions only have weak peaks corresponding to the highly stable $r_1$ structure, but are otherwise quite random, demonstrating the strength of the $r_1$ configuration, as the ensemble consists of the $r_1$ configuration and random ones. Taken together, Bcl-xL has a wide variety of conformations of its P2 site, including many closed conformations that cannot bind helical Bim. Furthermore, the high variability of the α3-α4 region suggests that the pocket is a highly flexible region. Given that the site around P4 is much more consistent and thus stable, it is clear that it is much easier for Bim to bind there first, as this site is in most cases already sufficiently formed for Bim to bind. Then, as the pocket opens up more, Bim is able to fold itself into a helical state and then bind to P2.

## Discussion

Since we have performed McMD simulations of Bim in isolation, as well as in the presence of Bcl-xL, we also wanted to compare the helicity of the peptide between the two conditions (using the same Amber99SB-ILDN force field). Fig. S12 compares the reweighted probabilities at 300 K of the helicity of the peptide (calculated via DSSP[43]) for both Bim in isolation (black) and Bim in the presence of Bcl-xL (red). Although the distributions have a similar shape with a peak near Gln60b/Glu61b, a slightly lower one at Glu68b and a valley around Arg64b/Ile65b, the helical probabilities of Bim in the presence of Bcl-xL are much higher. Notably, while the BH3 motif of Bim (Leu62b-X-X-X-Gly66b-Asp67b) in the IDR exhibits a low helical propensity in isolation, it shows a very high helical propensity in the presence of Bcl-xL, demonstrating that the presence of Bcl-xL considerably stabilizes the helical conformation. This also shows that our simulations reproduced both the IDR properties of Bim in isolation, as well as a strong helical conformation in the presence of Bcl-xL.

Our results indicated that the ff99SB-ILDN force field with the 3-point TIP3P water model could reproduce both disordered

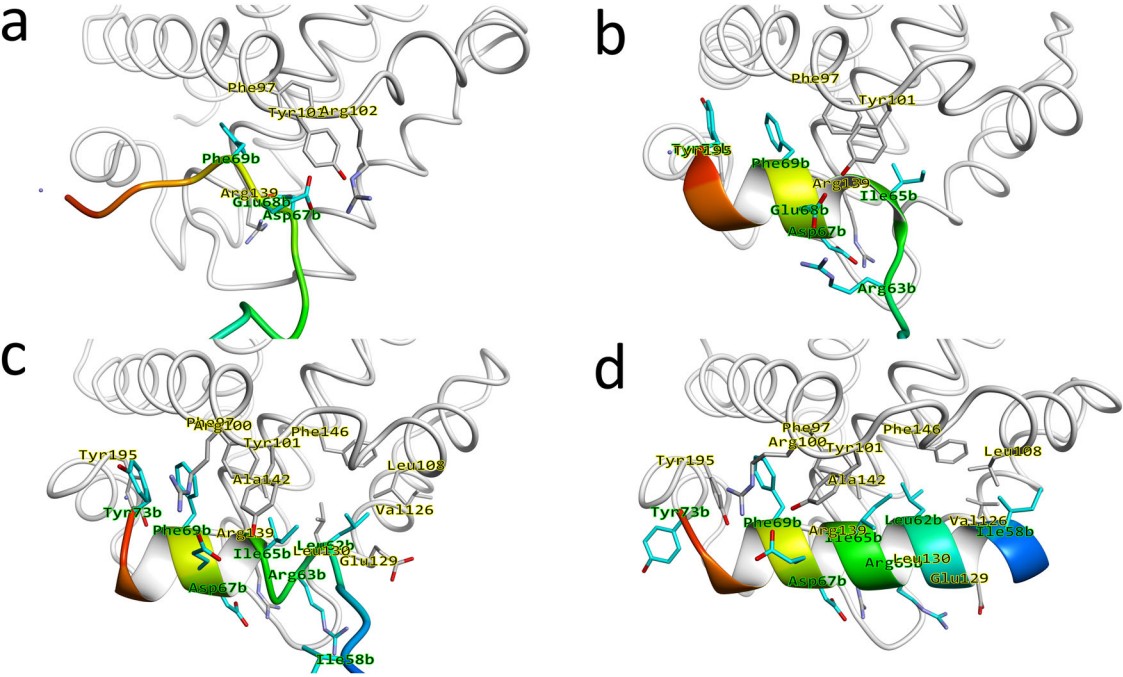

**Fig. 3 Milestone structures that define the binding mechanism between Bcl-xL and Bim. a** The structure at $\lambda = 10$ Å. **b** The structure at $\lambda = 8$ Å. **c** The structure at $\lambda = 5$ Å. **d** The structure at $\lambda = 0$ Å. In the figure, Bcl-xL is shown as a white tube model, while Bim is shown as a blue-red gradient colored cartoon model. Furthermore, important residues from Bcl-xL and Bim are shown as white and cyan stick models, respectively, and indicated in the figure in yellow and green, respectively.

characteristics of Bim in isolation (Fig. S3b), as well as Bim's ordered characteristics when in the presence of Bcl-xL (Fig. 2), where its helical propensity was greatly affected by the presence of Bim's partner protein (Fig. S12). Shabane et al, however, demonstrated that TIP3P waters for a set of IDPs (N-terminal zinc-binding domain of HIV-1 integrase, amyloid β-peptide and the H4 histone tail) did not reproduce the intrinsic characteristics well with canonical MD simulations, with TIP3P resulting in more compact structures[44]. On the other hand, the use of the 4-point OPC water model[45], resulted in wider and less compact conformational ensemble[44], suggesting that TIP3P waters stabilize peptides too much compared to OPC waters. However, our results (Fig. S3b) indicated that the TIP3P waters did not exert such an effect during our McMD simulations. To investigate the effect of OPC waters, we repeated our conformational sampling of the Bim peptide, replacing the (3-point) TIP3P waters with (4-point) OPC waters, keeping all other parameters the same. The resulting conformational ensemble (Fig. S13) is strikingly similar to that using TIP3P waters (Fig. S3b), although the number of structures within 2.5 kcal/mol is slightly higher for the OPC model (Table S8), suggesting that the choice of the water model only had a minimal effect. Therefore, because the use of a 4-point force field did not lead to a more disordered ensemble as suggested by previous work[44], and instead resulted in a ~33% increase in the system size (and thus a ~33% decrease in performance), using a more expensive 4-point force field like OPC does not seem like an efficient trade-off, at least when using McMD simulations. Granted, here we only looked at the Bim peptide, so it might be interesting to the IDP community to do a more extensive water force field comparison in a future work, with a wide range of IDPs using McMD simulations to get a more comprehensive idea of the effect of water models on the FEL and conformations of IDPs.

To analyze the binding mechanism in greater detail, Fig. 3 shows four milestone configurations at $\lambda = 10$ Å, 8 Å, 5 Å and 0 Å (see Fig. 1 for a definition of $\lambda$). Binding is initiated (Fig. 3a) on Bim's C-terminal side, where Phe69b approaches the P4 site of Bcl-xL. Glu68b—Arg139 as well as Asp67b—Arg102 form non-native salt bridges, preventing Phe69b from fully entering the P4 site. This early binding configuration consists of many non-native interactions, but Phe69b already makes native interactions with Phe97 and Tyr101 in this encounter configuration. Subsequently (Fig. 3b), once the salt-bridge between Asp67b and Arg102 is broken, the tension on Bim is released and Phe69b moves into the P4 site, while at the same time Asp67b is free to form its native interaction with Arg139, pushing Glu68b out of the way. As Phe69b moves into the P4 site, the site starts to assemble, with Tyr73b and Tyr195 also forming an interaction. Combined, this results in the C-terminal side of Bim folding into an α-helix up to the end of the BH3-motif (Asp67b), while the rest of the BH3-motif (Leu62-Gly66) is still in an unfolded conformation. Here, Glu68b faces outwards and interacts (non-natively) with Arg63b instead. Ile65b is located between sites P3 and P2, as Tyr101 partially blocks the P3 binding site due to hydrogen bonds it makes with the backbone of Bim. Also, at this point, the α3-α4 helices are still in a closed state, preventing Bim from binding to the P2 and P1 states. Next (Fig. 3c), presumably due to the strong pressure of the BH3-motif to form an α-helix, Tyr101 moves out of the way and the non-native interaction between Glu68b and Arg63b breaks. This allows Ile65b to move into the P3 site, elongating the helical conformation of Bim, as Leu62b starts to enter the P2 site. However, as the α3-α4 helices still haven't fully opened up, this prevents access of Bim to the P1 site. Finally (Fig. 3d), Tyr142 and Ile65b change their rotameric state, and a strong hydrophobic interaction between Leu108 and Val126 is formed, forming the P1 site. This allows Bim to fold completely into an α-helical conformation as Leu62b fully enters the P2 site and Ile58b enters the P1 site, completing the binding between Bim and Bcl-xL. In summary, Bim binds Bcl-xL via a two-step binding process; Bim first forms the initial encounter configuration with Bcl-xL (Fig. 3a). Then, via mutual induced-fit binding where their conformations adapt to each other (Fig. 3b/c), they form the native configuration (Fig. 3d).

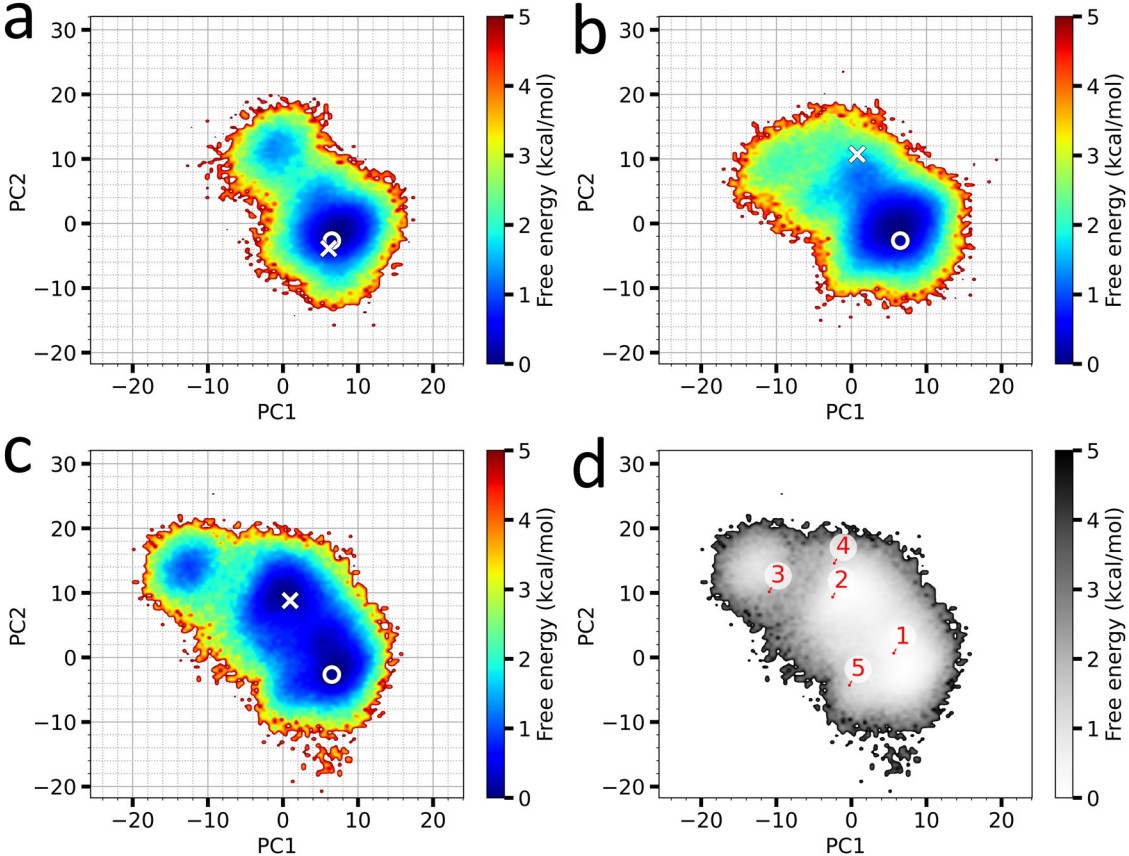

**Fig. 4 Free energy landscape of the Bcl-xL receptor protein in the presence of different ligands.** The landscapes are shown along the first and the second principal components (PC1 and PC2, respectively), obtained from each system's reweighted (300 K) multicanonical ensemble, with the contour value in kcal/mol. **a** Bcl-xL conformational landscape in the presence of WEHI-539. **b** Bcl-xL conformational landscape in the presence of ABT-737. **c** Bcl-xL conformational landscape in the presence of Bim. **d** Location of representative structures $r_k^{bcl}$ within 2.5 kcal/mol on the Bcl-xL landscape of **c**. The principal component analysis of Bcl-xL was performed based on the distance matrix calculated from atom pairs constructed from two groups, excluding neighboring residues ($i \pm 4$). The first group is defined as the Cα atoms from residues Ala84 to Tyr195, while the second group is defined as the Cα atoms from the residues Ala89, Glu124, Gly147 and Trp181. PC1 and PC2 contribute 15.58% and 9.34%, respectively, to the conformational variance of Bcl-xL. Also indicated in **a–c** are the apo conformation (initial structure used for the simulations) as a circle and the corresponding holo conformation as a cross.

We previously also performed McMD-based dynamic docking between Bcl-xL and two medium-sized compounds, WEHI-539 and ABT-737 (Fig. S14)[29]. To analyze the impact of the different ligands (WEHI-539, ABT-737 and Bim) on the conformation of Bcl-xL, we performed PCA on the Cα-atoms of the Bcl-xL structures using our three dynamic docking simulations. Figure 4 shows the 2D FEL of the Bcl-xL conformations for WEHI-539, ABT-737 and Bim. The image also shows the location of the experimental WEHI-539 (PDB ID 3ZLR)[17], ABT-737 (PDB ID 2YXJ)[46] and Bim (PDB ID 4QVF)[15], in addition to the apo (PDB ID 1R2D)[47] structure. Interestingly, the three FELs of Bcl-xL are different to one another, given that they were started from the same conformation (i.e., the apo-structure), showing that different ligands induce different conformations of Bcl-xL. Comparing the structure ensembles of Bcl-xL (Fig. 4a–c), some observations can be made; the area of the FELs increases, suggesting an increasingly wider conformational space of Bcl-xL comparing WEHI-539, ABT-737 and Bim, while the location of the stable basins also differ between the three systems. The apo and WEHI-539 bound experimental structures, both which have a closed cryptic pocket, are located nearby the large basin shared by all three systems, while the ABT-737 and Bim experimental structures are located nearby the second basin with an open cryptic pocket. To investigate the conformational preferences of these states, we applied our picking method on the Bcl-xL

conformations in the presence of Bim (i.e., using the data shown in Fig. 4c) to pick representative structures $r_k^{bcl}$ from the FEL, with the location of these structures on the FEL shown in Fig. 4d. Closely inspecting the structures corresponding to these conformations $r_k^{bcl}$ (Fig. 5), we can observe that PC1 appears to be correlated to the movement of helix α8, where higher PC1 values correspond to structures with α8 closer to the binding site and lower values to structures with α8 farther from the binding site. On the other hand, PC2 appears to be correlated with the collapse of the α3 helix, where lower PC2 values correspond to a more closed state and higher values to a more open state. Here, $r_1^{bcl}$ and $r_5^{bcl}$ correspond to closed states, while $r_2^{bcl}$, $r_3^{bcl}$ and $r_4^{bcl}$ correspond to open states. These results again demonstrate the flexibility of α3-α4, where sites P1 and P2 are located, similar to our analysis of the binding site (Fig. S10 and Table S7), which also showed that α3-α4 is highly flexible. In summary, the cryptic binding site of Bcl-xL (α3-α4) is a highly flexible region, where each ligand induces different conformational preferences upon Bcl-xL, with ABT-737 and Bim inducing an open state, while WEHI-539 primarily induces a closed conformation.

Given the above described effect of the various ligands on the conformational ensemble of Bcl-xL, some insight to the development of new drugs can be attained. The conformational preference of unbound Bcl-xL seems to prefer a structure with a closed cryptic (P1/P2) binding site, similar to the experimental

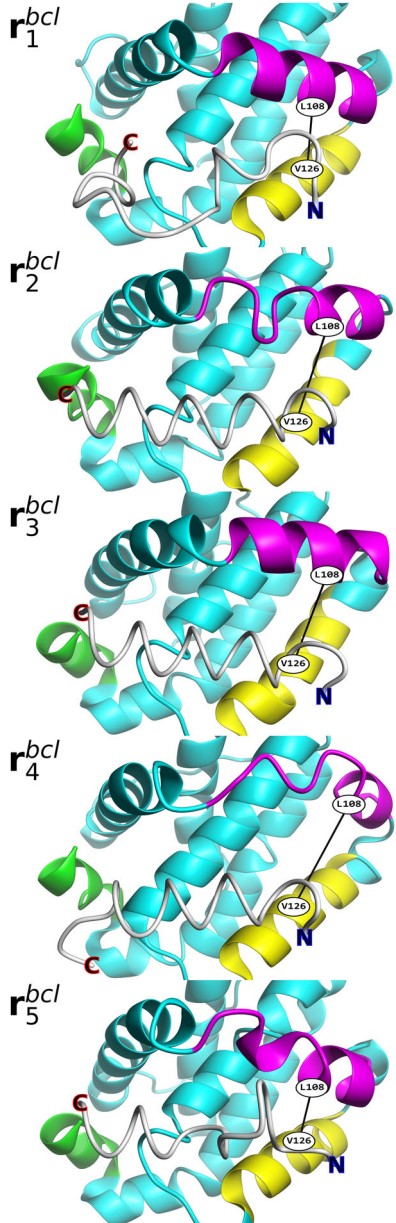

**Fig. 5 Conformations of Bcl-xL in the presence of Bim as shown on the FEL in Fig. 4.** Shown are the structures $r_k^{bcl}$ of Bcl-xL in the presence of Bim (white). The α helices α3, α4 and α8 of Bcl-xL are indicated in magenta, yellow and green, respectively, while the residues corresponding to d1, L108 and V126, also indicated, with the values corresponding to this distance listed in Table S7. Also indicated are the location of the N- and C-termini of Bim.

WEHI-539 and apo structures, while Bim induces an open Bcl-xL conformation. Since ABT-737 is a compound that was designed to mimic the Bcl-2 homology 3 only (BH3-only) motif as found in Bim, it makes sense that ABT-737 induces a similar effect on Bcl-xL as Bim does. However, the conformational flexibility of α3-α4 observed in our present simulations shows that it is a highly flexible region (Fig. S10). Previously we also observed that the binding configuration of WEHI-539 corresponding to the experimental structure to only be at the third rank[29]. Here, the substructure that tucks under α3 in the third-ranking and experimental structures, was instead facing towards the bulk along α4 in the top-ranking structure[29], indicating that the configuration observed in the experimental structure is more difficult

to attain, while also resulting in a higher ratio of closed Bcl-xL conformations, as the P2 site wasn't populated in that configuration. Combined, these observations suggest that the conformational flexibility of α3-α4 limit the rate at which a ligand (either a natural or designed compounds) can bind. Therefore, to increase the rate at which designed compounds could bind, it would be worthwhile to not specifically target the native bound structure ($\mathbf{r}_1$, $r_2^{bcl}$ and $r_3^{bcl}$), but the closed meta-stable structures observed in our present simulations, such as $\mathbf{r}_2$ ($r_1^{bcl}$) and $\mathbf{r}_6$, where Bcl-xL is in a closed conformation. In particular, $\mathbf{r}_6$ (Fig. S11), demonstrated a very high stability during our validation (Table 2), with a large number of residues being as stable as the native configuration in terms of their intermolecular contacts (Table S5). If a compound were to be designed that could effectively bind directly to these more predominant closed conformations of Bcl-xL (i.e., without requiring conformational changes of Bcl-xL to fully bind as observed in Bim, ABT-737 and WEHI-539), a faster association constant can most likely be obtained, leading to a higher affinity drug. To also ensure a high specificity, similar docking simulations could be performed with homologues of Bcl-xL, such as Bcl-2 and Mcl-1, to further optimize the compound to have Bcl-xL specific traits.

Figure 6 shows a schematic overview of the binding process. Here, along with the figure, a summary is given of the binding process, and the residues involved, which are summarized in Table 3. Initially, only the P4 site of Bcl-xL is accessible, which is where Phe69b initially binds, interacting with Phe97 and Tyr101, while the other sites (P3, P2, P1) are still inaccessible. As Bim binds, it folds into a helical conformation, while at the same time inducing a conformational change upon Bcl-xL, to slowly open more of the binding site. Ile65b subsequently binds to the P3 site, interacting with Tyr101, while Tyr73b stacks with Phe69b and Tyr195. In addition, an auxiliary interaction is formed between Asp67b and Arg139. Next, Bim initiates binding to P2, where the core of the hydrophobic interactions is formed, with Leu62b binding to the P2 pocket formed by Ala142, Tyr101, Phe97, Phe105 and Leu130, while also interacting with Ile65b. In addition, an auxiliary interaction is formed between Glu68b and Arg100. Finally, Bim binds to P1, where Ile58b interacts with Leu108, but also Leu62b, completing the mutual induced-fit binding process.

In summary, we have performed McMD-based dynamic docking simulations between Bcl-xL and Bim treating both the cryptic binding site of Bcl-xL and the conformation of Bim as flexible, and reproduced the native bound configuration. We reproduced the IDR properties of Bim in isolation, as well as the strong helical preference in the presence of Bcl-xL. Our binding pathway analysis showed that binding is initiated at the C-terminal of Bim (near site P4 of Bcl-xL), forming a low-affinity encounter configuration, and as Bim induces a conformational change of Bcl-xL to open up the P2 site, Bim folds into an α-helical conformation and binds Bcl-xL in the high affinity native configuration. This is also corroborated by our analysis of the pocket of Bcl-xL, which showed a high stability for the P4 site, with a much lower stability for the P2 and P1 sites near α3 and α4, where the N-terminal side of Bim binds. Finally, combined with the helical conformation analysis of both Bim in isolation as well as in the presence of Bcl-xL, our results suggest that binding is initiated at the C-terminal side of Bim to the stable P4 site as an encounter complex with Bim largely disordered. Then via mutual induced-fit binding, Bcl-xL and Bim adapt to each other to form the native binding configuration. Our results give insight into the binding mechanism of Bim, which can be used to design higher affinity inhibitor drugs to fight various cancers. Finally, it also provides insight into how other BH3-motifs might bind to Bcl-xL.

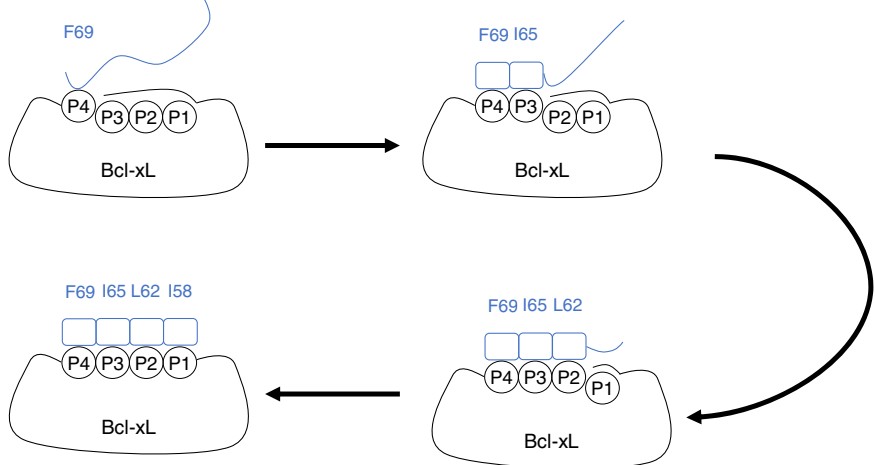

**Fig. 6 Schematic overview of mutual induced-fit binding between Bcl-xL (black) and Bim (blue).** The cryptic pocket consists of Sites P1–P3, where these sites tend to be inaccessible without a ligand bound. As Bim binds, it folds into a helical conformation (indicated by the boxes) and causes the cryptic pocket to open up and the sites to become accessible, allowing Bim to also bind to these sites.

**Table 3 Overview of residues on the surface between Bcl-xL and Bim and their role.**

| Residue | Role |
|---|---|
| Ile58b | Binds to site P1 |
| Leu62b | Binds to site P2 |
| Ile65b | Binds to site P3 |
| Glu68b | Interacts with Arg100, forms an interaction with Arg63b in an intermediary binding configuration |
| Asp67b | Interacts with Arg139, forms an interaction with Arg102 in intermediary binding configuration |
| Phe69b | Binds to the site P4 |
| Tyr73b | Interacts weakly with the P4 site |
| Phe97 | Member of the P2/P3 site |
| Arg100 | Interacts with Glu68b |
| Tyr101 | Member of the P3 site |
| Phe105 | Member of the P2 site |
| Leu108 | Member of the P1 site |
| Arg139 | Interacts with Asp67b |
| Leu130 | Member of the P1 site |
| Ala142 | Member of the P2 site |
| Tyr195 | Member of the P4 site |

## Methods

**Computational systems**. We used the same initial structure of Bcl-xL as in our previous work[29]. The apo structure of Bcl-xL in its monomeric state with PDB ID 1R2D (resolution 1.95 Å) was obtained from Protein Data Bank Japan (PDBj)[48–50]. The missing Ser28-Ile81 loop region was replaced by an 8-mer Gly linker using MODELLER[51]. The structure of Bim was taken from the holo complex structure of 4QVF and was cut to span only from Ile56b to Tyr73b (Ace-IWIA-QELRRIGDEFNAYY-Nhe), where the peptide was capped with Ace (CH₃CO-) and Nhe (-NH₂) residues, and was initially placed in the binding site. The structure was then rotated so that the plane corresponding to the binding site was perpendicular to the x-axis and a box of the size 110 × 62.5 × 62.5 Å was created around the complex and Bim was translated 40 Å along the x-axis into the bulk region. The box was solvated, with Na⁺ and Cl⁻ added to neutralize the system and bring the concentration to physiological levels (0.1 M). The Amber99SB-ILDN force field[52], monovalent ion parameters[53] and TIP3P[54] were used to parameterize the Bcl-xL and Bim, ions, and water molecules, respectively. The final system consisted of 2307 Bcl-xL atoms, 319 Bim atoms, 13141 water molecules, 30 Na ions and 24 Cl ions.

Gromacs 2021.3[55] was used to prepare and perform the simulations, which we have modified to perform the McMD-based dynamic docking simulations (see Notes). NVT (constant Number of particles, Volume and Temperature) or canonical MD simulations were performed at 300 K using the Bussi thermostat[56], while the NPT (constant Number of particles, Pressure and Temperature) or isobaric MD simulations, additionally used the C-rescale barostat[57] under 1 bar at 300 K. The long-range electrostatics were calculated using the zero-dipole summation method, which is a cutoff-based approach utilizing a well-defined pairwise function[26,58,59], with the damping factor α set to 0 Å⁻¹ and the atom-based cutoff length set to 12 Å. A time-step of 2 fs was used, with LINCS[60] to constrain the bond lengths and SETTLE[61] to constrain the water geometries. Energy minimizations, followed by 100 ps NVT and NPT simulations with position restraints on the heavy solute atoms were used to prepare the system. Table S9 summarizes these MD simulation parameters.

**Dynamic docking**. To prevent unfolding of Bcl-xL at high temperatures during the McMD simulations and the initial high temperature dissociation simulation, we employed the same distance restraints as for our previous simulations of Bcl-xL and two medium-sized ligands[29]. There, we used distance restraints between the backbone oxygen and nitrogen atoms for the residues that form hydrogen bonds that stabilize the secondary structure (in the case of Bcl-xL, only α-helices), with a flat-bottom region spanning between 0–4.5 Å, after which a 10 kcal/mol/Å² force constant is used to restrain the hydrogen bond. In addition, we had also added restraints between 9 Cα atom pairs (Arg91-Asp11, Leu13-Gly147, Leu13-Ala167, Glu133-His177, Tyr195-Ala89, Pro116-Val161, Ser2-Asn175, Val135-Trp181). These were restrained using a flat-bottom potential at ± 2 Å from their measured distance in the apo structure (PDB ID 1R2D[47]) using a 10 kcal/mol/Å² force constant. In addition to these previously used restraints, we also add one intramolecular distance restraint for Bim between its N-terminal and C-terminal residues. In particular, a 35 Å flat-bottom restraint between the Ace55b CH3 and Tyr73b Cα atoms (initial distance in the folded state is ~26 Å) was added to prevent extended forms from appearing that would not fit properly inside the simulation box. Finally, the translation and rotation of the center of mass (COM) of the protein was also restrained to keep Bcl-xL centered inside the box during the simulations using our COM restraint protocol, as before[29].

While we previously performed an exhaustive search of the entire configuration space between Bcl-xL and two medium-sized ligands[29], here we chose to restrain Bim to only sample the unbound region and the binding site. Figure 1c shows the box where the red subsection indicates the region to which the COM of Bim was restrained to stay inside. We used the McMD algorithm described in Note S1 on both of the systems, with 30 parallel trajectories initialized with different random seeds for the initial velocity from the initial structure constructed as described in section 1 of the Methods. After the initialization, first a 1 ns simulation at $T_{high}$ (700 K) was performed for each parallel trajectory to randomize the ensemble with the above described weak distance restraints. From here, the initial bias was estimated using Eq. S3 and subsequent iterations of increasing simulation lengths were executed, updating the bias using Eq. S4 between iterations, until a sufficiently flat potential energy distribution had been obtained corresponding to a wide $T_{mc}$ range of $T_{low} - T_{high}$, where $T_{low} = 280$ K. In total, the pre-run lasted for 24 μs (800 ns per trajectory) per system (Table S3). Finally, a 30 μs (1 μs per trajectory) production run was executed to sample the structures including bound and unbound states, which were saved at 5 ps intervals, producing 6.0 ×10⁶ structures.

**Dynamic docking analysis**. We used the same analyses techniques as we did in our recent works[28–31,41]. First, we performed PCA on a distance array derived from the structures. The array consists of intramolecular pairs of Bim and Bcl-xL and intermolecular pairs between Bim and Bcl-xL. For the intermolecular pairs, Cα atoms from Bcl-xL (from Ala149) versus the Cα atoms from Bim were taken. For the Bim intramolecular pairs, the Cα atoms plus a single representative atom from

the sidechain (except Gly) were taken, excluding pairs of residues i ± 3. For the Bcl-xL intramolecular pairs, the distances between the Cα atoms of Tyr101–Leu112 (α3) versus that of Ala149 (buried residue in α5) were calculated to include the flexibility of α3, as we did before[29]. The distance based approach does not require prior superposition of the structures unlike the quasi-harmonic approach[35,62], while taking periodic boundary conditions into account and being a more sensitive approach to detecting intermolecular contacts along the entire surface between the interacting molecules. The structures are then projected onto the first two principal components, and the probability of each bin $i$ on the landscape is calculated as $P_i = \sum_j P_c(E_j, 300\,K)$ using each structure $j$ within bin $i$. The free energy as the Potential of Mean Force (PMF) is finally calculated as $PMF_i = -RT \ln P_i$ for each bin (where $R$ is the gas constant), giving the 2D FEL after normalizing its minimum to zero and applying a cutoff at 5 kcal/mol.

After the PCA, we performed K-means clustering on the data, using $k' = 1000$ clusters and a set of PC dimensions, so that the sum of the contribution to their variance exceeds 90%, corresponding to PC1-PC17. For each cluster, one representative structure was then selected and the clusters were then ranked based on the relative free energy at 300 K giving the cluster free energy (CFE), which was calculated as $CFE_{k'} = -RT \ln P_{k'}$. Here, the summation probability, $P_{k'} = \sum_j P_c(E_j, 300\,K)$, was calculated using the reweighted probability $P_c$ of each structure $j$ corresponding to cluster $k'$. Then, using these representative structures, direct analysis on the structural similarity is performed using $R$-value analysis[27,42]. The $R$-value quantifies the degree at which intra- and/or inter-molecular contacts are preserved with respect to a reference structure, where in the case of analysis of binding configurations, the focus would be on intermolecular contacts between receptor and ligand. Starting from the most stable cluster $k' = 1$ in order of their free energy contribution, representative structures with an $R$-value > 0.7 were merged together. This gave us $k$ number of clusters and their corresponding representative structures $\mathbf{r}_k$, for each system. After calculating the free energy contribution of the merged clusters, we used a CFE cutoff value of 2.5 kcal/mol to distinguish between potentially interesting structures and less stable structures.

Representative complex structures $\mathbf{r}_k$ with a CFE within 2.5 kcal/mol that were obtained from the multicanonical ensemble, were further refined using canonical (NVT) MD simulations at 300 K without the restraints that were used during the McMD simulations (see section 2). For each representative structure, ten 100 ns MD simulations at 300 K were performed (with different random seeds for the initial velocity). Then, refined complex structures $\mathbf{q}_k$ were picked by taking the nearest-to-average structure from the final 40 ns of the canonical MD simulations. In addition, 400 K canonical MD simulations were performed to compare the relative stabilities of the binding configurations, like we have done before[23,27–31]. Finally, the trajectories were analyzed, where the average R($\mathbf{r}_k$)-value over the final 40 ns of the ten parallel trajectories at each temperature was calculated.

**Binding pathway analysis**. To analyze the binding mechanism, we picked connected binding representative structures along $\lambda$, which corresponds to the $x$-axis, starting from the equilibrated bound structure of $\mathbf{q}_1$ using the method we have described and used previously[23,27,29,30]. In short, the reaction coordinate is split into pre-defined windows (Table S6). In each window, a representative structure is picked from the multicanonical ensemble that is similar in terms of their $R$-value to the picked structure from the previous window. This produces a smoothly connected pathway of structures along the dissociation direction $\bar{\lambda}$ starting from the structure $\mathbf{q}_1$ to the unbound state.

**Reporting summary**. Further information on research design is available in the Nature Portfolio Reporting Summary linked to this article.

## Data availability

The representative structures and interactive versions of Figs. 1, 2c, d, 3, 5, S5, S6, S7, S8, S9 and S11 and the raw simulation data have been submitted to the Biological Structure Model Archive (BSM-Arc)[63], under BSM-00036 (https://doi.org/10.51093/bsm-00036).

## Code availability

The source code for executing McMD dynamic docking simulations, including a modified version of Gromacs and the analysis scripts, are available at https://gitlab.com/gjbekker/gromacs.

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

## Acknowledgements
This work was supported by Japan Agency for Medical Research and Development (AMED) to N.K., and by the Grand-in-Aid for Scientific Research from the Japan Society for the Promotion of Science to N.K. and G.-J.B. (JP20H03229). It was performed in part under the Cooperative Research Program of the Institute for Protein Research, Osaka University, CR-22-05. Computational resources of the TSUBAME3.0 system, Tokyo Institute of Technology, were provided by the HPCI Research Project (hp200011, hp200025, hp200063, hp210002, hp210005, hp210048, hp220002, hp220015 and hp220026).

## Author contributions
G.-J.B. and N.K. designed the study and performed the simulations; G.-J.B., M.A., K.O., Y.O. and N.K. analyzed the results; G.-J.B. and N.K. wrote the paper. All the authors discussed the research, edited the paper and approved its final version.

## Competing interests
The authors declare no competing interests.
