## [Peer Review File · Communications Biology]

Reviewers' comments:

Reviewer #1 (Remarks to the Author):

In this manuscript, the authors showed the dynamic behavior of the BIM binds to the Bcl-xL protein. The paper was technically sound and will help the researcher understand the Bcl-xL complex's binding pathway.

Before considering publication, authors should include the below comments in the manuscript.

1. How the helical structure of bim is formed needs to be clarified. Whether the structure formation during dynamic simulation in the presence of Bcl-xL? Authors should include these in the manuscript.
2. The authors did not explain the binding energy between the bim and the Bcl-xL protein throughout the manuscript. They should provide a table or graphs that show how binding energy varies through the simulation.

Reviewer #2 (Remarks to the Author):

Bekker et al reported the study of intrinsically disordered region of Bim binds to the flexible cryptic pocket of Bcl-xL. This study is strongly related to the effort fighting cancer disease. The authors applied their in-house developed method, so-called Multicanonical Molecular Dynamics-based docking methods. The authors provided a thorough insight into the binding mechanism of the complex including stable/meta-stable binding state, binding pathway.

The manuscript is well-constructed and easy to reach to people in the field.

To make the manuscript more attractive and thorough, some of the comments below are strongly suggested to justify in the manuscript.

1. The forcefield used in the simulations (AMBER99SB-ILDN) is not thoroughly describing the properties of IDR/IDP as reported in Robustelli et al, PNAS 115, E4758 (2018). This strongly affects the configurations of Bim itself on-the-way of recognition of IDR of Bim on the Bcl-xL. At least, the authors should show the binding mechanism unchanged with the change of water models (from TIP3P used in current simulations to OPC or TIP4P-D) by conventional MD of the transient states to the final binding state. This is because of in multiple publications in literature showed that the better water models, the better IDR/IDP can reproduce.
2. As shown in the simulation box in Fig 1c, the peptide IDR of Bim's length (in extended state) seems to be longer than the distance from alpha4 to the upper edge plus distance from alpha8 to the lower edge. This means if the IDR of Bim binds to alpha4, it can tentatively interact with alpha8 in its image. Therefore, authors should prove that the box is large enough for all the binding configurations of the complex.
3. Although the thorough analysis of interactions between Bim and Bcl-xL have been given, it is very difficult to understand from the manuscript which is the key-interactions between Bim and Bcl-xL during the process of folding upon binding. In break-down, in the initial state of contacts, which residues guide the Bim to the correct binding site? Which residues help the Bim to stay at correct binding site, which factors assist the Bim to fold at the correct binding site?
4. This is optional, but the authors provide the link to Github in which there is a modified version of GROMACS. However, the README file does not thoroughly change to reflect a modified version but only keeps it as original versions from GROMACS's developers. In addition, it is suggested to make a code patch for GROMACS to be able to run with McMD, until the method is decided to implement to the main branch of the GROMACS source.

Reviewer #3 (Remarks to the Author):

The apoptotic pathway mentioned in the introduction section will be easier to follow with a schematic of the pathway.

AMBER ff99SB-ILDN force field was reported to be more accurately predicting the structural ensemble, is there any particular reason for this? One of the main improvements made in AMBER ff99SB-ILDN force field was optimisation of the side-chain torsion potentials. Can this be related to the presence of amino acids with large side chains (like PHE, TYR, TRP in this case)?

PCA was performed on C- α distances, however for protein structures it is often important to include intramolecular features like dihedral torsional angles, intra-peptide H-bond to understand their dynamics.

In Fig-1b if the identified residues are coloured differently and/or is identified with a different drawing style it will be easier to follow.

Reviewer #1 (Remarks to the Author):

In this manuscript, the authors showed the dynamic behavior of the BIM binds to the Bcl-xL protein. The paper was technically sound and will help the researcher understand the Bcl-xL complex's binding pathway.

Before considering publication, authors should include the below comments in the manuscript.

1. How the helical structure of bim is formed needs to be clarified. Whether the structure formation during dynamic simulation in the presence of Bcl-xL? Authors should include these in the manuscript.

Although this is described throughout the manuscript, we have added a new scheme (Fig. 6, see below the text) that explains the binding mechanism, including the formation of the helical structure of Bim in the presence of Bcl-xL. In addition, we added the following paragraph summarizing the binding process to the discussion section:

Fig. 6 shows a schematic overview of the binding process. Here, along with the figure, a summary is given of the binding process, and the residues involved, which are summarized in Table 3. Initially, only the P4 site of Bcl-xL is accessible, which is where Phe69b initially binds, interacting with Phe97 and Tyr101, while the other sites (P3, P2, P1) are still inaccessible. As Bim binds, it folds into a helical conformation, while at the same time inducing a conformational change upon Bcl-xL, to slowly open more of the binding site. Ile65b subsequently binds to the P3 site, interacting with Tyr101, while Tyr73b stacks with Phe69b and Tyr195. In addition, an auxiliary interaction is formed between Asp67b and Arg139. Next, Bim initiates binding to P2, where the core of the hydrophobic interactions is formed, with Leu62b binding to the P2 pocket formed by Ala142, Tyr101, Phe97, Phe105 and Leu130, while also interacting with Ile65b. In addition, an auxiliary interaction is formed between Glu68b and Arg100. Finally, Bim binds to P1, where Ile58b interacts with Leu108, but also Leu62b, completing the mutual induced-fit binding process.

2. The authors did not explain the binding energy between the bim and the Bcl-xL protein throughout the manuscript. They should provide a table or graphs that show how binding energy varies through the simulation.

We use the cluster free energy (CFE), which describes the relative free energy with respect to the cluster with the largest population, to describe the free energy differences between the various configurations, where Table 1 lists the CFE using all the data. We have added a new table (Table S6) that describes the change in our results when only using the first 25%, 50% and 75% of the simulation data when analyzing the results (PCA, clustering and representative picking). Although the picked representative changes, the top-ranking structure is consistently the near-native structure identified in Table 1. We also refer to this new Table S6 from the main text (new addition in bold):

Fig. S4 shows the structures for these representative configurations, where Table 1 lists several statistics of them, where these structures are ranked by the free energy contribution of the corresponding cluster. **In addition, analysis results using 25 %, 50 % and 75 % of the simulations from the production run are shown in Table S6. Here, the top-ranking structure in each subsection is consistently the native configuration, indicating that the docking simulations have quickly converged.**

Reviewer #2 (Remarks to the Author):

Bekker et al reported the study of intrinsically disordered region of Bim binds to the flexible cryptic pocket of Bcl-xL. This study is strongly related to the effort fighting cancer disease. The authors applied their in-house developed method, so-called Multicanonical Molecular Dynamics-based docking methods. The authors provided a thorough insight into the binding mechanism of the complex including stable/meta-stable binding state, binding pathway. The manuscript is well-constructed and easy to reach to people in the field. To make the manuscript more attractive and thorough, some of the comments below are strongly suggested to justify in the manuscript.

1. The forcefield used in the simulations (AMBER99SB-ILDN) is not thoroughly describing the properties of IDR/IDP as reported in Robustelli et al, PNAS 115, E4758 (2018). This strongly affects the configurations of Bim itself on-the-way of recognition of IDR of Bim on the Bcl-xL. At least, the authors should show the binding mechanism unchanged with the change of water models (from TIP3P used in current simulations to OPC or TIP4P-D) by conventional MD of the transient states to the final binding state. This is because of in multiple publications in literature showed that the better water models, the better IDR/IDP can reproduce.

Based on the literature (Robustelli et al 2018, but also Shabane et al 2019), it looks like these 4-point water force fields increase the dynamics by introducing an additional entropic effect onto the protein/peptide molecule during conventional canonical MD simulations. Here, we are performing Multicanonical MD simulations, where the

enhanced effect on the dynamics is much, much stronger than the comparatively minor entropic effect that a change in water model would provide (as we are effectively simulating in a temperature range corresponding to 280 K to 700 K). Thus, the effect of changing the water force field would be expected to be minimal during an McMD simulation. On the other hands, changing from a 3-point force field to a 4-point force field, would considerably decrease the performance of the simulation (as most of the particles in the system correspond to water molecules). Still, to validate this effect on Bim, we have built a system where we replaced the 3-point force field (TIP3P waters) with a 4-point force field (OPC waters) and re-executed the conformational sampling of Bim using Multicanonical MD, which we had used to validate and choose the force field to use for the docking simulations. After analyzing the results, we found that with the 2D FEL (Fig. S13) is similar to that of the 2D FEL (Fig. S2b) obtained from the simulation with TIP3P (with independently performed PCAs), although the number of representative structures within a 2.5 kcal/mol cutoff with the OPC force field (Table S7) was larger than the number of representative structures with the TIP3P force field (Table S2), although at this point we're only talking about 0.01 % differences in cluster populations. Notable though is that neither ensemble reproduced the helical structure found in the experimental complex structure with Bcl-xL. In summary, the use of OPC waters did not increase the disordered nature of the Bim peptide beyond what we observed with TIP3P, while increasing the number of particles (from N=19499 to N=26088), thus decreasing performance.

We have also added the following paragraph to the Discussion section:

Our results indicated that the ff99SB-ILDN force field with the 3-point TIP3P water model could reproduce both disordered characteristics of Bim in isolation (Fig. S2b), as well as Bim's ordered characteristics when in the presence of Bcl-xL (Fig. 2), where its helical propensity was greatly affected by the presence of Bim's partner protein (Fig. S11). Shabane et al, however, demonstrated that TIP3P waters for a set of IDPs (N-terminal zinc-binding domain of HIV-1 integrase, amyloid β -peptide and the H4 histone tail) did not reproduce the intrinsic characteristics well with canonical MD simulations, with TIP3P resulting in more compact structures.⁴⁴ On the other hand, the use of the 4-point OPC water model,⁴⁵ resulted in wider and less compact conformational ensemble,⁴⁴ suggesting that TIP3P waters stabilize peptides too much compared to OPC waters. However, our results (Fig. S2b) indicated that the TIP3P waters did not exert such an effect during our McMD simulations. To investigate the effect of OPC waters, we repeated our conformational sampling of the Bim peptide, replacing the (3-point) TIP3P waters with (4-point) OPC waters, keeping all other parameters the same. The resulting conformational ensemble (Fig. S13) is strikingly similar to that using TIP3P waters (Fig. S2b), although the number of structures within 2.5 kcal/mol is slightly higher for the OPC model (Table S7), suggesting that the choice of the water model only had a minimal effect. Therefore, because the use of a 4-point force field did not lead to a more disordered ensemble as suggested by previous work,⁴⁴ and instead resulted in a ~ 33% increase in the system size (and thus a ~ 33 % decrease in performance), using a more expensive 4-point force field like OPC does not seem like an efficient trade-off, at least when using McMD simulations. Granted, here we only looked at the Bim peptide, so it might be

interesting to the IDP community to do a more extensive water force field comparison in a future work, with a wide range of IDPs using McMD simulations to get a more comprehensive idea of the effect of water models on the FEL and conformations of IDPs.

2. As shown in the simulation box in Fig 1c, the peptide IDR of Bim's length (in extended state) seems to be longer than the distance from alpha4 to the upper edge plus distance from alpha8 to the lower edge. This means if the IDR of Bim binds to alpha4, it can tentatively interact with alpha8 in its image. Therefore, authors should prove that the box is large enough for all the binding configurations of the complex.

Yes, the box size is technically not large enough, however, due to the nature of the McMD simulations where unstable binding configurations only appear at very high temperature and that partially bound structures are less stable than fully bound ones, such structures are only expected to be rare occurrences, and even then, only at high temperature. To validate this, we scanned all the structures for configurations where the termini of Bim interacted with $\alpha 4$ and $\alpha 8$ at the same time, while crossing the PBC. We found a number of these structures, and after reweighing these structures to room temperature (300 K), we found that these structures have a log-population of approximately -4162.58, which corresponds to a population of zero. Thus, the occurrence of these PBC-crossing configurations, although not desired, has no measurable effect on the analysis, and thus do not influence any of our original observations or conclusions.

3. Although the thorough analysis of interactions between Bim and Bcl-xL have been given, it is very difficult to understand from the manuscript which is the key-interactions between Bim and Bcl-xL during the process of folding upon binding. In break-down, in the initial state of contacts, which residues guide the Bim to the correct binding site? Which residues help the Bim to stay at correct binding site, which factors assist the Bim to fold at the correct binding site?

We have added a new Table 3 that lists the interacting residues from both Bcl-xL and Bim, along with their role in the binding. Also, we have added a new scheme (Fig. 6, see below the text) that explains the binding mechanism, including formation of the helical structure of Bim in the presence of Bcl-xL. In addition, we added the following paragraph summarizing the binding process to the discussion section:

Fig. 6 shows a schematic overview of the binding process. Here, along with the figure, a summary is given of the binding process, and the residues involved, which are summarized in Table 3. Initially, only the P4 site of Bcl-xL is accessible, which is where Phe69b initially binds, interacting with Phe97 and Tyr101, while the other sites (P3, P2, P1) are still inaccessible. As Bim binds, it folds into a helical conformation, while at the same time inducing a conformational change upon Bcl-xL, to slowly open more of the binding site. Ile65b subsequently binds to the P3 site, interacting with Tyr101, while Tyr73b stacks with Phe69b and Tyr195. In addition, an auxiliary interaction is formed between Asp67b and Arg139. Next, Bim initiates binding to P2, where the core of the hydrophobic interactions is formed, with Leu62b binding to the P2 pocket formed by Ala142, Tyr101, Phe97, Phe105

and Leu130, while also interacting with Ile65b. In addition, an auxiliary interaction is formed between Glu68b and Arg100. Finally, Bim binds to P1, where Ile58b interacts with Leu108, but also Leu62b, completing the mutual induced-fit binding process.

4. This is optional, but the authors provide the link to Github in which there is a modified version of GROMACS. However, the README file does not thoroughly change to reflect a modified version but only keeps it as original versions from GROMACS's developers. In addition, it is suggested to make a code patch for GROMACS to be able to run with McMD, until the method is decided to implement to the main branch of the GROMACS source.

We have made some changes to the README file to reflect the changes made to Gromacs. In addition, we have included instructions on how to build and use the modified version of Gromacs. On the other hand, we have decided to not provide a code patch, as this will likely not be maintained very well, unlike our Git repository, which is regularly maintained (it's kept fairly up-to-date with the main Gromacs repository, although main releases generally tend to wait until after the first point-release). In addition, by using Git, it is easy for users to maintain an up-to-date version of Gromacs, and installing Gromacs is not necessary (so they can even have multiple Gromacs versions, if desired).

Reviewer #3 (Remarks to the Author):

The apoptotic pathway mentioned in the introduction section will be easier to follow with a schematic of the pathway.

We have added Scheme S1 (see below), which describes the role of Bim and Bcl-xL in the apoptotic pathway, to the Supplementary Information.

AMBER ff99SB-ILDN force field was reported to be more accurately predicting the structural ensemble, is there any particular reason for this? One of the main improvements made in AMBER ff99SB-ILDN force field was optimisation of the side-chain torsion potentials. Can this be related to the presence of amino acids with large side chains (like PHE, TYR, TRP in this case)?

Important here is that we compare ff99SB-ILDN to ff14SB, which are two completely different force fields. The authors of ff14SB compared their force field to the old ff99SB (i.e., non-ILDN) force field (using conventional canonical MD simulations), and showed an increase in helical propensity of ff14SB for their test systems (two designed peptides). Notably, we didn't execute any simulations with the old ff99SB force field, as that was not the goal of our work; we simply wanted to pick a force field (from the set of force fields we have used in the past) that demonstrated that it could reproduce the disordered characteristics of the Bim peptide when executing McMD simulations, which the ff99SB-ILDN force field did reproduce, while the ff14SB force field did not. Although we did not test it, the side chain parameter changes might still influence the backbone dihedral propensities when looking at the free energies, which would become apparent when doing, e.g., Multicanonical MD simulations.

PCA was performed on C- α distances, however for protein structures it is often important to

include intramolecular features like dihedral torsional angles, intra-peptide H-bond to understand their dynamics.

First, for docking, the inter-molecular distances are more important than intra-molecular distances. Secondly, by using the C α atom distances, we still indirectly measure the effects of backbone hydrogen bonding. Instead of directly measuring specific bb-O and bb-N pairs, we measure the distances of all C α pairs, indiscriminate of any reference conformation of Bim/Bcl-xL. It should be noted that our assumption is always “we don't know the correction conformation or binding configuration”, meaning we have to apply a generalized analysis, which is why we prefer to use C α distances. This can then capture most of the conformational (secondary structure formation) and configurational (binding) characteristics of the structures.

In Fig-1b if the identified residues are coloured differently and/or is identified with a different drawing style it will be easier to follow.

We have changed the colors of the labels to match that of Fig. 2, i.e. by using a green outline for the residues from Bim, while using a yellow outline for residues from Bcl-xL (Fig. 1b is shown below).

REVIEWERS' COMMENTS:

Reviewer #1 (Remarks to the Author):

All the comments were addressed and included in the manuscript. Now, I recommend this manuscript for further processing.

Reviewer #2 (Remarks to the Author):

The authors have addressed all the technical and scientific points carefully. The manuscript has been improved and this should be excitingly disclosed to the public as soon as possible.